

# The TROPOMI surface UV algorithm

Anders V. Lindfors[1], Jukka Kujanpää[1], Niilo Kalakoski[1], Anu Heikkilä[1], Kaisa Lakkala[1],
Tero Mielonen[1], Maarten Sneep[2], Nickolay A. Krotkov[3], Antti Arola[1], and Johanna Tamminen[1]

[1]Finnish Meteorological Institute, Finland
[2]Royal Netherlands Meteorological Institute (KNMI), The Netherlands
[3]NASA Goddard Space Flight Center, Maryland, USA

*Correspondence to:* Anders Lindfors (anders.lindfors@fmi.fi)

**Abstract.** The TROPOspheric Monitoring Instrument (TROPOMI) is the only payload of the Sentinel-5 Precursor (S5P), which is a polar orbiting satellite mission of the European Space Agency (ESA). TROPOMI is a nadir-viewing spectrometer measuring in the ultraviolet, visible, near-infrared and the shortwave infrared that provides near-global daily coverage. Among other things, TROPOMI measurements will be used for calculating the UV radiation reaching Earth's surface. Thus,

the TROPOMI Surface UV product will contribute to the need of monitoring UV radiation by providing daily information on the prevailing UV conditions over the globe. The TROPOMI UV algorithm builds on the heritage of the OMI (Ozone Monitoring Instrument) and AC SAF (Satellite Application Facility for Atmospheric Composition and UV Radiation) algorithms. This paper provides a description of the algorithm that will be used for estimating surface UV radiation from TROPOMI observations. The TROPOMI Surface UV product includes the following UV quantities: the UV irradiance at 305, 310, 324, and

380 nm; the erythemally weighted UV; the vitamin-D weighted UV. Each of these are available as (i) daily dose or daily accumulated irradiance, (ii) overpass dose rate or irradiance, and (iii) local noon dose rate or irradiance. In addition, all quantities are available corresponding to actual cloud conditions and as clear-sky values, corresponding to otherwise the same conditions but assuming a cloud-free atmosphere. This yields 36 UV parameters altogether. The TROPOMI UV algorithm has been tested using input based on OMI and GOME-2 (Global Ozone Monitoring Experiment–2) satellite measurements. These preliminary

results indicate that the algorithm is functioning according to expectations.

## 1   Introduction

Solar ultraviolet (UV) radiation has a broad range of effects concerning life on Earth. Because of its high photon energy, UV radiation influences human health, terrestrial and aquatic ecosystems, air quality, and materials in various ways. For a more detailed discussion on the different effects of UV radiation, see, e.g., UNEP (2015). In order to study and assess these

effects and their implications thoroughly, information is needed on the geographical and temporal distribution of UV radiation reaching Earth's surface.

After the discovery of severe ozone depletion in Antarctica during the austral spring (Farman et al., 1985), it was soon recognized that the stratospheric ozone content was declining also in the Arctic and at mid-latitudes (WMO, 1989). Subsequently, the UV radiation reaching Earth's surface increased during the last decades of the 20th century. The strongest increase took



place in the high latitudes of the southern hemisphere, but also the Arctic and mid-latitudes in both hemispheres have experienced UV increases. Outside the polar regions, the increase has been mostly around 5% or less compared to pre-industrial conditions, whereas at high and polar latitudes, where ozone depletion has been larger, increases have been more substantial (UNEP, 2011).

Thanks to the Montreal Protocol, concentrations of ozone depleting substances in the stratosphere are currently decreasing and the total ozone column is on a recovery path back towards pre-1980 levels, which are expected to be reached before the middle of the 21st century (UNEP, 2011). Here, it is worthwhile noting that other factors than the ozone column, for example, clouds and surface albedo, also play an important role in determining how much UV radiation reaches Earth's surface. Current projections of the future UV radiation climate indeed bring forth the complex connections between climate change, strato-
spheric ozone depletion, and stratospheric dynamics, which all influence the UV radiation of our future climate (Williamson et al., 2014). Thus, it is of great importance to continue monitoring the UV radiation reaching Earth's surface.

The Sentinel-5 Precursor (S5P) is a polar orbiting satellite mission planned to be launched during the second half of 2017. The only payload of the mission is the TROPOMI instrument (The TROPOspheric Monitoring Instrument), which is a nadir-viewing push-broom spectrometer measuring in the ultraviolet, visible, near-infrared and the shortwave infrared. The S5P
mission will have a sun-synchronous orbit with an ascending node equatorial crossing at 13:30, which in conjunction with a wide swath of 2600 km provides near-global daily coverage. Among other things, TROPOMI measurements will be used for calculating the UV radiation reaching Earth's surface. Thus, the TROPOMI Surface UV product will contribute to the need of monitoring UV radiation by providing daily information on the prevailing UV conditions over the globe.

This paper provides a description of the algorithm that will be used for estimating surface UV radiation from TROPOMI
observations, including a discussion on algorithm heritage and some example results based on currently active satellite instruments.

## 2  Heritage

The method of Eck et al. (1995) is one of the first satellite-based algorithms for estimating UV irradiances at Earth's surface presented in the literature. Their method is based on measurements of backscattered UV radiation by the TOMS (Total Ozone
Mapping Spectrometer) instrument. The general idea in their approach is to first calculate clear-sky UV irradiances, taking into account the total ozone column as retrieved from measurements by the same instrument, and then in a second step correct these clear-sky irradiances for the attenuation caused by clouds. The cloud effect is determined based on the Lambertian Equivalent Reflectance (LER) at 360 nm or 380 nm, which is a measure of how reflective the clouds are. The higher the reflectance, the thicker is the cloud (optically) and thus the stronger is the attenuation of the surface UV irradiance caused by this cloud. This
relationship has later been refined (Krotkov et al., 2001) to also take into account the effect of multiple scattering between the surface and the cloud (see also Herman et al., 2009), yielding the expression:

$$C_T = (1 - LER)/(1 - R_G) \tag{1}$$



where $R_G$ is the LER value representing the surface (that is, LER at cloud-free conditions), and $C_T$ is the cloud correction (Tanskanen et al., 2006) or cloud transmittance (Krotkov et al., 2001; Herman et al., 2009) factor, which is also commonly referred to as the Cloud Modification Factor (CMF) (Calbo et al., 2005; den Outer et al., 2005; Lindfors et al., 2007). This factor is applied to the clear-sky UV irradiance ($E_{clear}$) to obtain an estimate of the UV irradiance under the cloud (E):

$$E = E_{clear} \cdot CMF = E_{clear} \cdot C_T \tag{2}$$

Many satellite UV algorithms presented after the pioneering work of Eck et al. (1995) bear heritage to their method. However, it is worth noting that in the TOMS UV algorithm, Eck's LER method was replaced with a more realistic plane parallel cloud model at the turn of the century (Krotkov et al., 1998, 2001). In the plane parallel cloud model, the cloud optical depth is estimated using radiative transfer calculations assuming a plane parallel water cloud, and given as input the measured
reflectance. The advantage of this approach is that it describes more realistically the wavelength-dependent attenuation of incoming solar radiation by clouds (Krotkov et al., 2001; Lindfors and Arola, 2008) and also that it accounts for the directional distribution of radiation reflected off the cloud.

The Dutch-Finnish Ozone Monitoring Instrument (OMI) onboard NASA's Aura satellite launched in 2004 continues the TOMS UV record to present date. The OMI UV algorithm (Tanskanen et al., 2006) is, in essence, very similar to that of TOMS.
A recent update of the OMI UV algorithm added climatological aerosol information in order to account for the attenuation caused by absorbing aerosols (Arola et al., 2009).

For estimating daily UV exposure, or daily doses, a disadvantage of the above-described TOMS branch of satellite UV algorithms is that they are based on only one satellite overpass per day. This means that the algorithm does not account for variations in the cloudiness that occur within the day. Instead the cloud situation of the overpass is assumed to be valid for the
whole day. Therefore, days with varying cloud conditions cause a larger uncertainty in the daily UV doses (Bugliaro et al., 2006).

Another approach for estimating surface UV irradiances from satellite measurements is to use cloud information from geostationary satellites in combination with total ozone column from polar orbiting platforms. This approach has the advantage of having almost continuous cloud observations, but, on the other hand, is not able to cover high-latitudes because of challenging
view angles. Geostationary satellite UV algorithms have been developed for both the European and North American regions (Verdebout, 2000; Gadhavi et al., 2008).

In order to gain a better picture of the intra day cloud variability, some algorithms utilize measurements from multiple polar orbiting satellites (Matthijsen et al., 2000; Ciren and Li, 2003; Lindfors et al., 2009). Such an approach is used also in the Offline UV (OUV) product (Kujanpää and Kalakoski, 2015) provided by EUMETSAT's AC SAF (Satellite Application Facility for
Atmospheric Composition and UV Radiation; formerly known as the Ozone and Atmospheric Chemistry Monitoring Satellite Application Facility). The OUV algorithm utilizes cloud measurements by the AVHRR (Advanced Very High Resolution Radiometer) instruments carried by two different polar orbiting satellites (Metop and NOAA POES) to produce a global gridded daily UV product (Level 3 product). In the OUV algorithm, the total ozone column is taken from GOME-2.





As explained in more detail in Section 3, the approach chosen for estimating surface UV irradiances based on TROPOMI measurements builds on the TOMS-OMI heritage, while also utilizing parts of the AC SAF OUV algorithm. In practice, this means that the surface UV is estimated using radiative transfer calculations implemented in a Look-Up-Table (LUT) to keep computational demands at a reasonable level. The inputs for these radiative transfer calculations are essentially the TROPOMI-

retrieved total ozone column and reflectance at 354 nm together with information on the surface albedo and the atmospheric aerosol load taken from other sources. Here, the reflectance at 354 nm is used to determine the cloud optical depth.

Of the above-discussed heritage UV products, the TOMS UV record is the longest. The TOMS UV record is based on measurements by a series of satellites. It begins with measurements by the Nimbus 7 satellite in 1978 and ends with the Earth Probe in 2007. The OMI UV product covers the years from 2004 to present and is currently foreseen to last for at least a few

more years, while TROPOMI will continuate this satellite UV record further into the future. Thus, there will most probably be at least some years of overlap between TROPOMI and OMI. This overlap will be very useful for validation purposes and time series analyses. Although the AC SAF OUV comprises a somewhat different type of UV product (because it is a direct L3 product), it is worthwhile noting that the OUV record covers the period from 2007 onwards, and will thus also be available for comparison with the TROPOMI UV product.

## 15  3   TROPOMI surface UV algorithm

For the TROPOMI UV algorithm, we have chosen to use the VLIDORT radiative transfer model (Spurr, 2006). VLIDORT is a vector discrete ordinate radiative transfer model that accounts for polarization. It has been used, for example, in the satellite UV algorithm of AC SAF (Kujanpää and Kalakoski, 2015).

The inputs to the VLIDORT radiative transfer calculations are essentially the total ozone column ($\Omega$), the surface albedo

($\rho_s$), and the cloud optical depth ($\tau_c$). Based on these, the UV irradiances at Earth's surface are calculated through a LUT based approach, which reduces the computational demands as compared to online radiative transfer model calculations. Here, the total ozone column is a retrieved L2 product of TROPOMI, available in the Total Ozone product (Spurr et al., 2016), while the cloud optical depth is estimated as part of the UV algorithm (see section 3.3) based on the measured reflectance at 354 nm ($R_{354}$) provided by the L2 Aerosol Index (AI) product (Stein Zweers, 2016).

Figure 1 presents a schematic flow diagram of the TROPOMI UV algorithm. In practice, the TROPOMI UV algorithm uses two sets of LUT's instead of explicitly running the radiative transfer model. One LUT gives $\tau_c$ using $R_{354}$ and $\rho_s$ as main inputs, while the other one gives the UV irradiance using $\Omega$, $\tau_c$, and $\rho_s$ as main inputs. Here, $\rho_s$ is taken from a climatology created for the OUV algorithm. It uses the monthly Minimum Lambert Equivalent Reflectivity (MLER) climatology (Herman and Celarier, 1997) for regions and time periods with permanent or negligible snow/ice cover, while a climatology better

capturing the seasonal changes in the surface albedo (Tanskanen, 2004) during the transition periods is used elsewhere (see Kujanpää and Kalakoski (2015) for details). A correction for the effect of absorbing aerosols corresponding to climatological aerosol properties is applied, following the OMI aerosol correction approach (Arola et al., 2009).





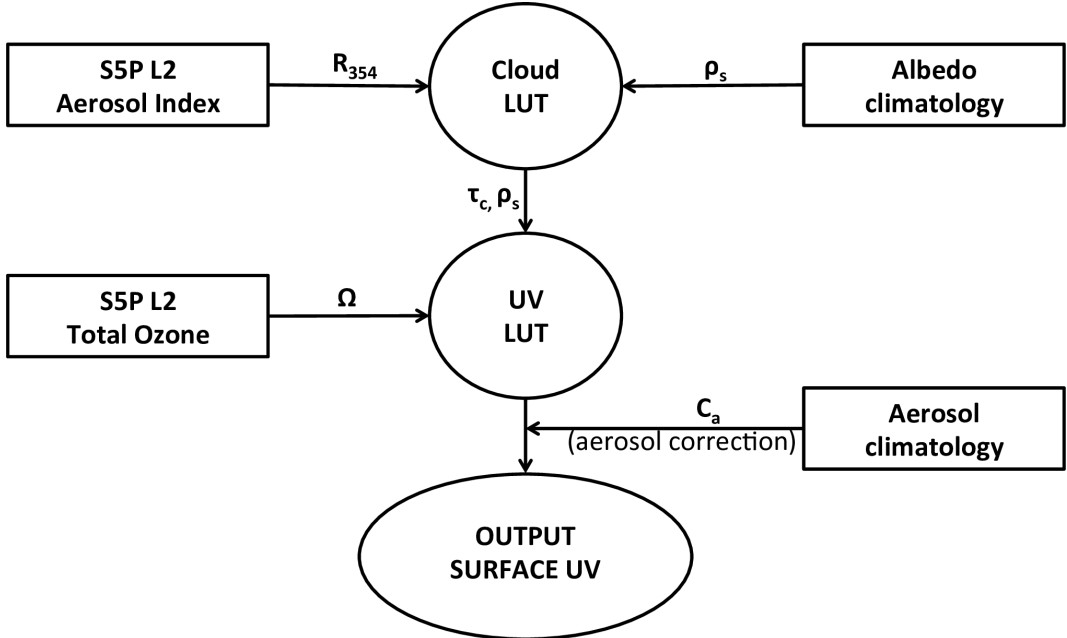

**Figure 1.** Schematic flow chart of the TROPOMI UV algorithm. See text for details.

In the following subsections, we first discuss briefly the influence various factors such as clouds and aerosols have on surface UV irradiances. Thereafter, we describe the approach used in the radiative transfer model within the TROPOMI UV algorithm, including a description of how the LUT's are implemented.

### 3.1 Factors affecting surface UV

The UV irradiance at Earth's surface is largely determined by the solar zenith angle, clouds, total ozone column, surface albedo, aerosols, Earth-Sun distance, and altitude or pressure. The different factors affecting surface UV radiation have been discussed in the literature (see, e.g., Weatherhead et al. (2005), and reference therein). Here, we include only a brief discussion on these factors and encourage the interested reader to study the literature more in-depth.

**Solar zenith angle.** The solar zenith angle ($\theta_0$) determines the optical path of the direct radiation component through the
atmosphere and is therefore the most important factor for the surface UV irradiance. Through its influence on the optical path, the solar zenith angle modifies the effect other factors have on the surface UV. The smaller the solar zenith angle (higher sun), the more UV radiation reaches the surface, and vice versa.

**Clouds.** Clouds attenuate UV radiation in a way which is similar to the familiar attenuation of visible radiation under cloudy skies; they reflect part of the incoming radiation back to space, thereby reducing the irradiance reaching Earth's surface.
Josefsson and Landelius (2000) found an average CMF (Cloud Modification Factor) of 0.4 for overcast conditions at a Swedish station, with a fairly large variation in the observed CMF values around this average, depending on the detailed properties of the



prevailing cloud. For cloud amounts less than 4/8 (octas), Josefsson and Landelius found a CMF larger than 0.9, indicating that UV radiation is only weakly attenuated as long as less than half of the sky is covered by clouds. den Outer et al. (2005) found a climatological yearly average CMF of 0.68 for erythemal UV in the Netherlands, while for global solar radiation (integrated over the wavelength range 300–3000 nm), they found a climatological CMF of 0.57. Thus, there is a wavelength dependence

in the attenuation of incoming solar radiation by clouds, with radiation of shorter wavelengths making it to the surface more effectively. The physical aspects of this effect have been discussed by Lindfors and Arola (2008).

In radiative transfer, the cloud optical depth ($\tau_c$) is often used as a measure of the opacity of the cloud. $\tau_c$ defines how much a beam of radiation passing through the cloud is attenuated, following the Beer–Lambert–Bouguer law.

Although clouds generally attenuate the UV radiation reaching the surface, it is worthile noting that in certain conditions,

they can also act to increase the surface UV irradiance to above what would be expected in otherwise equivalent but cloud-free conditions. When the sun is not obscured by clouds, they can act as reflecting surfaces enhancing the UV irradiance reaching the surface (e.g., Lubin and Frederick, 1991).

**Total ozone column.** The total ozone column ($\Omega$) is predominantly a measure of the ozone content of the stratosphere, because most of the ozone of the atmosphere resides there. As ozone strongly absorbs UV radiation at wavelengths below 320

nm, the stratospheric ozone layer protects life on Earth from too intense UV radiation. Variations in the total ozone column are reflected in the UV irradiances at the surface. The larger the total ozone column, the lower is the UV irradiance and vice versa. The exact relationship between total ozone and surface UV depends on the wavelength considered, and also on the total ozone column and the solar zenith angle. For erythemally weighted UV and moderate to high sun ($\theta_0 < 60°$), a 1% change in the total ozone column leads roughly to a 1–1.2% change in the UV (Weatherhead et al., 2005).

**Albedo.** The surface UV albedo ($\rho_s$) is generally low, being of the order of 0.05 for most surfaces, as shown, for example, by the TOMS satellite data (Herman and Celarier, 1997). Surfaces covered by snow or ice, however, exhibit much higher surface albedos (Blumthaler and Ambach, 1988; Feister and Grewe, 1995). A high surface albedo may enhance the UV level at the surface significantly due to multiple scattering between the surface and the atmosphere above. It has been shown, for example, that a surface covered by fresh and clean snow (albedo 0.8) enhances the UV irradiance at 320 nm by a factor of 1.5

as compared to snow-free, low albedo conditions (Lenoble, 1998).

The UV irradiance at a specific location is not only influenced by the local surface albedo at the measurement site, but also by the often inhomogeneous surface conditions of the surrounding area extending a few tens of kilometers away from the measurement site (Degünther et al., 1998; Degünther and Meerkötter, 2000). The effect of the varying surface conditions in the surroundings of the measurement site can be represented by a so-called effective albedo. The effective surface albedo can be

thought of as the albedo required in a one-dimensional radiative transfer model in order to produce a cloud-free UV irradiance that is in agreement with measurements.

**Aerosols.** Aerosols usually decrease the level of UV radiation reaching the surface, partly by scattering radiation back to space and partly by absorption. The aerosol load of the atmosphere varies strongly in time and space. Therefore, the effect of aerosols on the UV irradiance reaching the surface also shows strong variations. During a measurement campaign in Greece

in 1996, aerosols were found to decrease the UV irradiance by 5–35% as compared to aerosol-free conditions (Kylling et al.,





1998). These numbers are similar to the yearly average aerosol attenuation in various parts of the globe corresponding to the aerosol correction applied in the OMI UV algorthim (Arola et al., 2009). This correction is based on the aerosol absorption optical depth ($\tau_{aa}$), and will be used also in the TROPOMI UV algorithm (see section 3.5).

**Altitude.** The UV irradiance at the surface usually increases with increasing altitude ($z$) (Gröbner et al., 2000). This is due

to the fact that the surface pressure ($p_s$) is smaller at higher altitude. In other words, there is less atmosphere above at high altitudes, and therefore less scattering and absorption taking place. Moreover, tropospheric ozone is mostly located at low altitudes, which is also the case for other pollutants such as aerosols. At high altitudes there are also typically less clouds, or even clouds beneath acting as a reflecting surface with high albedo, thus increasing the UV irradiance compared to less elevated sites.

**Earth-Sun distance.** The Earth-Sun distance ($R$) varies over the course of the year because of the elliptic shape of Earth's orbit around the Sun. Because of this variation, the irradiance at the top of the atmosphere varies as a function of $1/\mathrm{R}^2$. This variation can be taken into account using a multiplicative factor to irradiances representing the baseline Earth-Sun distance of 1 AU.

## 3.2    General radiative transfer setup

The aim of the radiative transfer setup of the TROPOMI UV algorithm is to account for the factors influencing the surface UV irradiance discussed above. VLIDORT is a one-dimensional radiative transfer model where the vertical structure of the atmosphere is represented by 30 model layers, as depicted in Figure 2. At the very bottom, there is the surface, which in our setup is a Lambertian reflector characterized by its albedo. The layer 1–2 km above the surface (layer 2) includes a homogeneous water cloud. In addition, all layers include Rayleigh scattering by air molecules ($p$ is the pressure), absorption

by ozone, and temperature information.

The cloud (layer 2) consists of water droplets with a size distribution following the C1 model by Deirmendjian (1969) yeilding an effective cloud droplet radius of $\mathrm{r_{eff}} = 6\mu\mathrm{m}$. The same cloud model has been used in the TOMS/OMI and the AC SAF UV algorithms (Krotkov et al., 2001; Kujanpää and Kalakoski, 2015).

The temperature and the ozone density of each layer in the radiative transfer model is set according to the TOMS V7

climatology (Wellemeyer et al., 1997), which gives climatological ozone and temperature profiles for three broad latitude bands (low latitude, middle latitude, and high latitude) as a function of total ozone column.

This general setup of the VLIDORT model is used for both (i) calculating the cloud optical depth, and (ii) calculating surface UV irradiances, as described in the following subsections. All radiative transfer calculations were done using the extraterrestrial solar spectrum of OMI (Dobber et al., 2008) and assuming an Earth-Sun distance of 1 AU.







**Figure 2.** Schematic structure of the model atmosphere.

## 3.3 Cloud LUT

The main input used to estimate the cloud optical depth ($\tau_c$) is the reflectance at 354 nm ($R_{354}$) taken from the TROPOMI AI L2 output. $R_{354}$ is defined as (Stein Zweers, 2016, their eq. 4-1):

$$R_{354} = \frac{\pi I_{354}}{E_{0,354} cos(\theta_0)} \tag{3}$$

where $I_{354}$ is the radiance at 354 nm reflected by Earth (atmosphere and surface) measured by TROPOMI and $E_{0,354}$ is the solar irradiance at 354 nm at the top of the atmosphere, and $\theta_0$ is the solar zenith angle. $E_{0,354}$ is measured by TROPOMI on a daily basis, and is corrected for the Doppler-shift in the measured spectrum due to the relative motion of the satellite with respect to the Sun. Both $I_{354}$ and $E_{0,354}$ are the average over 5 consecutive spectral pixels centered at 354 nm of the TROPOMI instrument. Furthermore, they have both been normalized to correspond to an Earth-Sun distance of 1 AU.

In our radiative transfer calculations, the combined spectral response of these 5 consecutive pixels are represented by a 0.88 nm wide (corresponding to 5 pixels 0.22 nm apart), flat-top response curve with a half-gaussian with a Half-Width-Half-Maximum of 0.27 nm at each end.

The approach here is somewhat different from that used in the OMI UV algorithm, where the radiance of a single spectral pixel at 360 nm is used to determine the reflectance, from which the cloud optical depth is consequtively calculated. We believe the approach chosen for TROPOMI will constitute an improvement because (i) averaging over 5 channels will help reduce noise in the signal, and (ii) while there is some $O_2 - O_2$ absorption at 360 nm (Thalman and Volkamer, 2013) that may influence the estimated cloud optical depth, 354 nm is free of such absorption features.



**Table 1.** Node points of the cloud optical depth look-up table.

| parameter | acronym | unit | node values |
|---|---|---|---|
| solar zenith angle | $\theta_0$ | degree | 0, 5, 10,..., 80 |
| viewing zenith angle | $\theta_v$ | degree | 0, 5, 10,..., 70 |
| relative azimuth angle | $\phi_r$ | degree | 0, 20, 40,..., 180 |
| cloud optical depth | $\tau_c$ | – | 0, 0.5, 1.0, 2.0, 4.0, 8.0, 16.0, 32.0, 64.0, 128.0, 256.0, 500.0 |
| surface albedo | $\rho_s$ | – | 0, 0.3, 0.6, 1.0 |
| surface pressure | $p_s$ | atm | 0.7, 1.0 |

The cloud LUT was calculated using the VLIDORT radiative transfer model. $R_{354}$ was calculated by systematically varying the cloud optical depth ($\tau_c$), the solar zenith angle ($\theta_0$), the viewing zenith angle ($\theta_v$), the relative azimuth angle ($\phi_r$) between the sun and the satellite, the surface albedo ($\rho_s$), and the surface pressure ($p_s$). The outcome is a look-up-table spanning all relevant combinations of these paramaters that gives $\tau_c$ as a function of the other parameters. Here, $\tau_c$ is an effective optical

depth in the sense that it corresponds to the cloud optical depth of a homogeneous water cloud, as specified above, that produces the best match with the measured $R_{354}$ given the other input parameters. Note that the ozone content of the atmosphere has a negligible influence on $R_{354}$. Therefore, we used a constant ozone profile in these calculations, corresponding to the middle latitude TOMS V7 profile with a total ozone column of 325 DU (i.e., M325 in Table 2).

The nodes of the cloud LUT are listed in Table 1. All inputs needed to retrieve $\tau_c$ are based on the TROPOMI AI L2 output,

except the albedo, which is based on a climatology as explained above. $p_s$ of the AI L2 output is normally based on the pressure fields of the ECMWF (European Centre for Medium-Range Weather Forecasts). In case of missing data from the ECMWF, $p_s$ is calculated from the altitude above sea level using the hydrostatic equation and assuming a scale height of 8.3 km. The altitude in the AI L2 output is based on Global Multi-resolution Terrain Elevation Data 2010 (Danielson and Gesch, 2011).

To demonstrate the idea of estimating $\tau_c$ based on the measured $R_{354}$, Figure 3 shows $\tau_c$ as a function of $R_{354}$ for a low

albedo ($\rho_s = 0.04$) and a fairly high albedo ($\rho_s = 0.60$) case. For the low albedo case, $R_{354}$ increases strongly with increasing $\tau_c$ up to optical depths of around 20, whereafter the increase gradually levels off as $R_{354}$ saturates towards large $\tau_c$. Note that for the high albedo case, $R_{354}$ is less sensitive to changes in $\tau_c$ than for the low albedo case. This means, in practice, that it is more difficult to accurately estimate $\tau_c$ for high albedo cases. Finally, the figure also brings forth the strong Rayleigh scattering of radiation in the UV; for cloudless skies ($\tau_c = 0$), the low albedo $R_{354}$ is around 0.25.

## 3.4 UV LUT

The solar irradiance at Earth's surface is a function of wavelength. Furthermore, the effect of the incoming radiation on, for example, a specific photochemical reaction or a biological response, depends on the wavelength. In order to provide a measure of the effectiveness of radiation as regards a specific effect, various action spectra (weighting functions) have been introduced





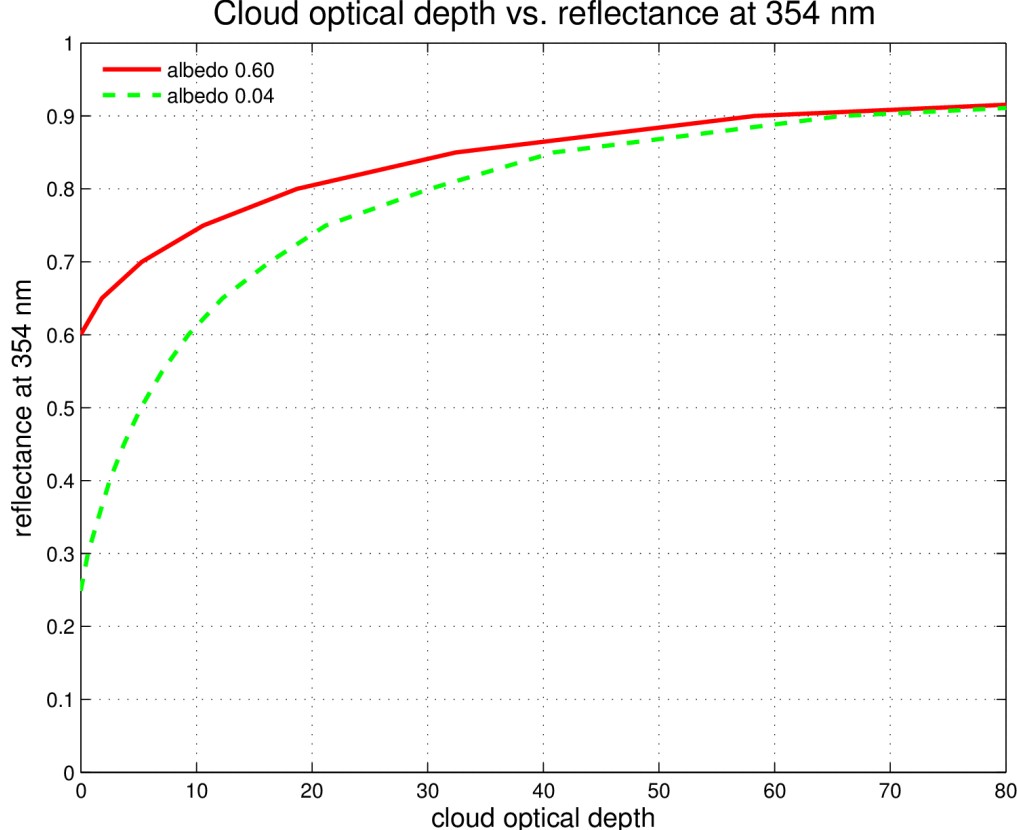

**Figure 3.** Cloud optical depth versus the reflectance at 354 nm according to the look-up-table. For the relationship shown here, the surface pressure was $p_s =$ 1 atm and the solar and viewing geometry was $\theta_v = 40°$, $\theta_0 = 45°$, $\phi_r = 60°$.

(e.g., Aphalo et al., 2012), that can be used to calculate effective spectral irradiances and dose rates (e.g., Kujanpää and Kalakoski, 2015, eq. 2).

The UV LUT of the TROPOMI algorithm includes UV irradiances at four selected wavelengths, namely 305, 310, 324, and 380 nm, and erythemally (Webb et al., 2011) and vitamin-D weighted UV dose rates (CIE, 2006, the tabulated data provided in the publication were linearly interpolated to obtain a complete action spectrum). Note that both the erythemal and the vitamin-D action spectrum gives large weight to radiation of wavelengths below 320 nm (see Fig. 3 in Kujanpää and Kalakoski, 2015).

Similarly to the cloud LUT, the LUT for UV irradiances and dose rates was created by running the radiative transfer model by systematically varying the following input paramters: $\theta_0$, $\Omega$, $\tau_c$, $\rho_s$, and $p_s$. Table 2 lists the nodes chosen for the UV LUT. Here, the ozone profile follows the TOMS V7 climatology as explained above.

For comparison with ground-based measurements of UV irradiances, it is worthwhile noting that all calculations for the UV LUT were done using a triangular 1 nm FWHM (Full-Width Half-Maximum) slit function, which is commonly used as





**Table 2.** Node points of the look-up table for the dose rates and UV irradiances at selected wavelengths. The full 26 profile set of the TOMS V7 climatology is used. L, M and H refer to the low, middle and high latitude profiles, respectively, while the numbers refer to total ozone columns in DU.

| parameter | acronym | unit | node values |
|---|---|---|---|
| solar zenith angle | $\theta_0$ | degree | 0, 5, 10,..., 85, 88 |
| TOMS V7 profiles | – | – | L225, L275,..., L475 |
| | | | M125, M175,...M575 |
| | | | H125, H175,...H575 |
| cloud optical depth | $\tau_c$ | – | 0, 0.39, 0.92, 1.7, 2.7, 4.1, 6.1, 8.9, 13, 18, |
| | | | 25, 36, 50, 70, 96, 130, 190, 260, 360, 500 |
| surface albedo | $\rho_s$ | – | 0, 0.1, 0.2,..., 1.0 |
| surface pressure | $p_s$ | atm | 0.7, 1.0 |

a standard slit function, for example, when comparing different ground-based spectrometers (Gröbner et al., 2005). Note also that the wavelengths of the extraterrestrial spectrum used in our radiative transfer calculations (Dobber et al., 2008) have been corrected to correspond to wavelengths in the atmosphere rather than in vacuum.

### 3.5 Daily cycle and post processing

The calculated UV quantities are corrected for the variation in the Earth-Sun distance (see section 3.1) and the attenuation caused by absorbing aerosols in a post processing step. After these corrections have been applied, the daily doses and daily accumulated irradiances are calculated by integrating over the 24 hour time window centered at local solar noon. For this, a time step of half an hour is used. It is worth empasizing that the TROPOMI UV algorithm does not account for variation in the cloud cover within the day, but instead assumes that the cloud optical depth inferred from the overpass measurement is valid
the whole day.

The absorbing aerosol correction follows the approach of Arola et al. (2009) used in the OMI UV algorithm. It is based on a monthly aerosol climatology by Kinne et al. (2013). The correction is a multiplicative factor ($C_a$) which depends on the aerosol absorption optical depth ($\tau_{aa}$):

$$C_a = \frac{1}{1 + 3\tau_{aa}} \tag{4}$$

Here, the factor 3 in the denominator represents average conditions according to earlier studies on the behavior of satellite-estimated UV as compared to ground-based measurements and its dependence on $\tau_{aa}$ (Krotkov et al., 2005; Arola et al., 2005). Arola et al. (2009) found a significantly reduced bias in the OMI UV product when compared to ground-based measurements over Europe after this correction had been applied.



As the aerosol absorption optical depth depends on wavelength, so does the aerosol correction factor ($C_a$) to be applied in the TROPOMI UV algorithm. For the UV irradiances at selected wavelengths (i.e, 305, 310, 324, and 380 nm), we apply a wavelength-specific aerosol correction. For the erythemally weighted and vitamin-D weighted UV irradiances, we apply the aerosol correction corresponding to 310 nm.

## 4  TROPOMI UV Product

### 4.1  Output and example results

The TROPOMI L2 UV Product includes the following UV quantities: the UV irradiance at 305, 310, 324, and 380 nm; the erythemally weighted UV; the vitamin-D weighted UV. Each of these are available as (i) daily dose or daily accumulated irradiance, (ii) overpass dose rate or irradiance, and (iii) local noon dose rate or irradiance. In addition, all quantities are available corresponding to actual cloud conditions and as clear-sky values, corresponding to otherwise the same conditions but assuming a cloud-free atmosphere. This makes 36 UV parameters altogether (see Table 3).

The L2 Aerosol Index and Total Ozone Column products used as input to the UV product are both based on measurements by TROPOMI's UVVIS spectrometer, band 3, which covers the wavelength range 320–405 nm (Loots et al., 2016). The ground resolution of these measurements, and thus also of the TROPOMI UV product, is 7 x 3.5 $\mathrm{km}^2$ in nadir, while the largest pixels throughout the swath are foreseen to be roughly 9 x 14 $\mathrm{km}^2$.

During the development of the TROPOMI UV algorithm, the UV processor software has been tested using GOME-2 and OMI satellite data as a surrogate for real TROPOMI measurements. The GOME-2 and OMI data have been processed using the TROPOMI Aerosol Index (Stein Zweers, 2016) and Total Ozone (Spurr et al., 2016) algorithms to produce realistic TROPOMI-like L2 output.

Figure 4 shows, as an example, the reflectance at 354 nm, the retrieved cloud optical depth, and the UV index of solar noon produced using OMI-based test data for 13 August 2007, processed to yield L2 Aerosol Index and Total Ozone output files that were given as input to the TROPOMI UV algorithm. The Finnish UV measurement stations Jokioinen (south-western Finland) and Sodankylä (northern Finland) are marked on the map. The OMI overpass time for both stations was close to 10:30 UTC and less than 15 minutes from local solar noon.

The figure shows relatively high reflectances ($R_{354}$) corresponding to cloudy areas over the central Baltic (east of Sweden) as well as over large parts of Sweden, Norway, Finland, and the Kola Peninsula. The cloud optical depth in these areas varies from values below 5 to values over 50. Over the western part of Russia, on the other hand, there is a rather cloud-free region extending to the south-eastern part of Finland. In this area, cloud optical depths are below 1.

The figure also shows the UV index (UVI) at local solar noon. The UVI has been introduced by the World Health Organization (WHO, 2002) as a tool to inform the public about UV radiation. Thus, we here use the UVI to demonstrate the example output of the TROPOMI UV algorithm. The UVI is formulated based on the erythemally weighted UV dose rate, scaled to reach a convenient number.



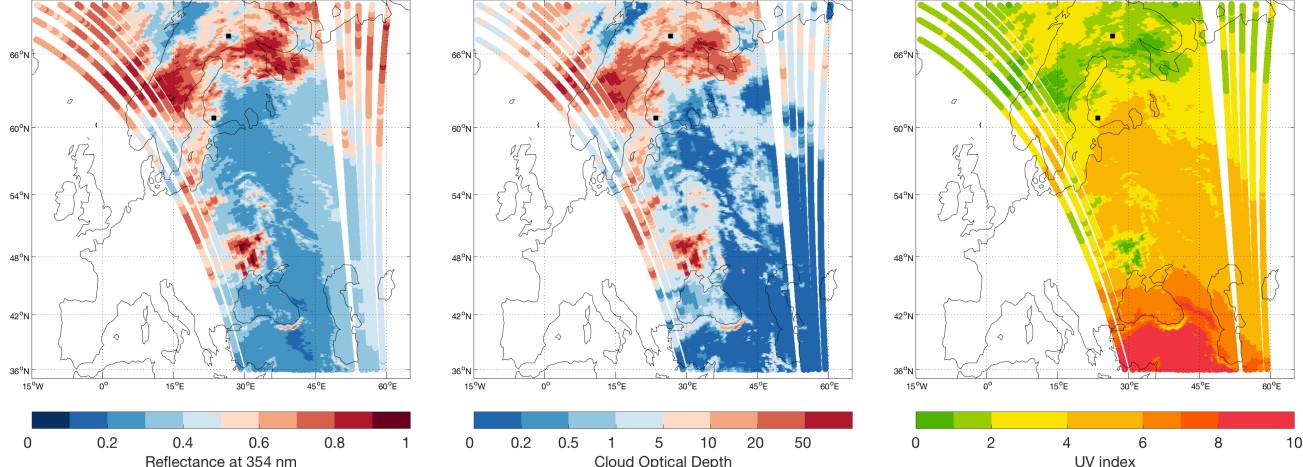

**Figure 4.** The reflectance at 354 nm (left panel), the cloud optical depth (middle panel), and the UV Index at solar noon (right panel) of the TROPOMI UV algorithm based on OMI test data for 13 August 2007. Jokioinen (south-western Finland) and Sodankylä (northern Finland) are marked with black squares.

The solar noon UVI of Fig. 4 reflects the aforementioned variations in the cloud optical depth. Also the noon solar zenith angle has a strong influence on the UVI. UVI < 2 prevail in the cloudy Scandinavian and North Atlantic areas, while UVI > 4 are present in the relatively cloud free areas of south-eastern Finland and western Russia. Going further south, cloud free areas with UVI > 8 can be seen south of the Black Sea, and also in the mountaineous regions between the Black Sea and the Caspian

Sea where an influence of the altitude can be observed.

  Figure 5 shows a more detailed comparison of the TROPOMI UV product with ground-based measurements by Brewer spectroradiometers (Lakkala et al., 2008) of the Finnish Meteorological Institute. Table 3 lists all 36 UV quantities of the L2 TROPOMI UV output for the satellite pixel closest to Sodankylä.

  The UVI measured by the Brewer spectrophotmeter in Sodankylä (Fig. 5) shows reduced values due to clouds during

the morning and afternoon hours, while the values around noon are close to those expected for cloud-free conditions. Also for Jokionen, the Brewer-measured UVI is close to expected values for cloud-free conditions, although the midday hours in general are somewhat cloudy. The very morning and late afternoon hours in Jokioinen also appear cloud-freee. The retreived effective cloud optical depths are 2.3 and 3.0, respectively, for Sodankylä and Jokioinen, indicating presence of a cloud over both stations during the satellite overpass. This is consitent with both the Brewer UV measurements and with supporting

satellite data (AVHRR data collected by the Finnish Meteorological Institute; not shown). Finally, Fig. 5 also demonstrates the effect of assuming that the overpass cloud optical depth is valid throughout the day: while the Brewer measurements indicate significant variability in the UV irradiance caused by clouds, the TROPOMI UV (red curve) is evenly attenuated throughout the day.



**Table 3.** TROPOMI UV output for Sodankylä 13 August 2007.

| | 305 nm | 310 nm | 324 nm | 380 nm | erythemal | vitamin-D |
|---|---|---|---|---|---|---|
| | $[\mathrm{mWm^{-2}nm^{-1}}]$ | | | | $[\mathrm{mWm^{-2}}]$ | |
| overpass | 16.98 | 44.03 | 178.91 | 374.14 | 74.34 | 135.73 |
| overpass clr | 20.52 | 52.93 | 215.35 | 473.35 | 89.95 | 163.64 |
| noon | 17.01 | 44.07 | 179.02 | 374.37 | 74.42 | 135.90 |
| noon clr | 20.55 | 52.98 | 215.48 | 473.59 | 90.04 | 163.83 |
| | $[\mathrm{Jm^{-2}nm^{-1}}]$ | | | | $[\mathrm{Jm^{-2}}]$ | |
| daily | 374 | 1123 | 5612 | 12204 | 1961 | 3293 |
| daily clr | 453 | 1358 | 6824 | 15958 | 2396 | 3989 |

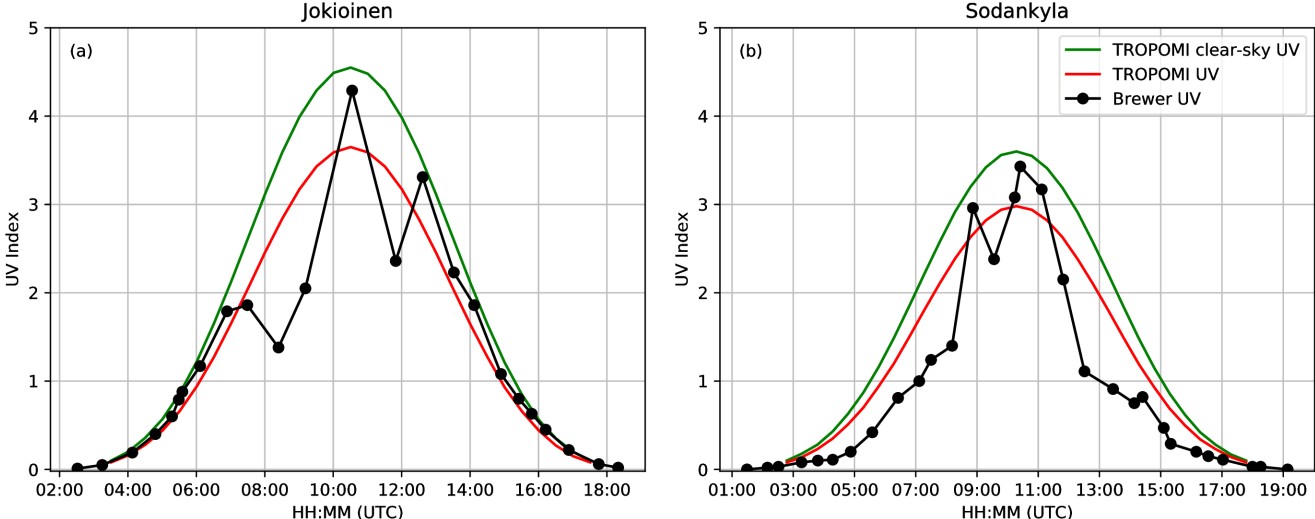

**Figure 5.** The UV Index as measured by a Brewer spectrophotometer in (a) Jokioinen, and (b) Sodankylä on 13 Aug 2007 together with results from the TROPOMI UV algorithm based on OMI measurements.

## 4.2 Expected uncertainty

The errors associated with the estimated UV quantities using the TROPOMI UV algorithm are expected to be similar to the errors of the heritage algorithms, namely the OMI surface UV product, which is most similar to that of TROPOMI. For TROPOMI, no validation results are yet available, while OMI, on the other hand, has been scrutinized against ground-based measurements in a number of studies presented in the literature.

For OMI, Tanskanen et al. (2007) summarized their validation results as follows:





For flat, snow-free regions with modest loadings of absorbing aerosols or trace gases, the OMI-derived daily erythemal doses have a median overestimation of 0–10%, and some 60 to 80% of the doses are within ±20% from the ground reference. For sites significantly affected by absorbing aerosols or trace gases one expects, and observes, bigger positive bias up to 50%. For high-latitude sites the satellite-derived doses are occasionally up to 50% too small because of unrealistically small climatological surface albedo.

After inclusion of the absorbing aerosol correction in the OMI algorithm, it is expected that the overestimation due to aerosols should have reduced considerably, which indeed was the case for the European stations included in the study of Arola et al. (2009). However, even after this correction, systematic overestimation of 20–30% remained for Rome (Italy) and Reading (England). Furthermore, it is worthwhile noting that the aerosol climatology represents typical or average conditions, which means that day to day variations in the aerosol load at polluted locations may still cause occasional strong overestimation of the surface UV.

A recent study by Bernhard et al. (2015) assessed errors in the OMI UV caused by inaccuracies in the surface UV albedo climatology used in the algorithm. Their results show that the OMI UV can have a bias exceeding 50% - in both negative and positive direction - when the OMI surface albedo is too high or too low. Note, however, that although the percentage error may seem large, the figures of Bernhard et al. (2015) correspond to conditions with relatively low sun and therefore the errors remain small on an absolute scale.

A more comprehensive uncertainty analysis of the TROPOMI UV product is planned, which will be based on uncertainties in the inputs given to the LUT's and how the uncertainties of all input parameters combined influence the estimated surface UV.

## 5 Conclusions

This paper describes the TROPOMI UV algorithm. The algorithm has been tested using realistic input, based on OMI and GOME-2 satellite measurements. These preliminary results indicate that the algorithm is functioning according to expectations, also when compared with ground-based Brewer spectrophotometer UV measurements in Finland. A proper evaluation of the performance of the TROPOMI UV product can, however, only be done after launch when real data start becoming available.

There is a strong need for continued monitoring of the UV radiation reaching Earth's surface. The TROPOMI UV record will build upon the heritage of satellite-retrieved surface UV that starts with TOMS in 1978, and continues to present date thanks on the OMI UV record (see, e.g., Ialongo et al., 2011).

*Author contributions.* A.V.L. designed the presented algorithm with help from all co-authors. A.V.L., J.K., and N.K. implemented the algorithm with help from N.A.K., M.S., A.A. and J.T. A.H., K.L. and T.M. helped to interpret the results. All co-authors participated in writing the manuscript.



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
