# Peer review of "The TROPOMI surface UV algorithm"

_Atmospheric Measurement Techniques, 2017_

## Referee Comment (RC1) · Anonymous Referee #1 · 18 Sep 2017

The TOMS and OMI instruments installed on a series of NASA satellites have been of utmost importance for monitoring changes in ozone and UV radiation from space. OMI became operational in 2004 and is now near the end of its lifetime. Because TROPOMI is one of the replacement instruments, it is important that the instrument's characteristics and the methods for processing raw data from the instrument are well defined so that the long time-series of ozone and UV data products from the legacy instruments can be continued without breaks in radiometric scale and accuracy. The manuscript by Lindfors et al. is important and should be published because it documents the algorithm that will be used to calculate UV dose rates and doses from TROPOMI observations. The manuscript is scientifically sound, well structured, and well written, and should be published with only minor changes. My main suggestion is to also include a short paragraph summarizing the difference between the UV algorithm currently used for the processing of OMI observations and the new algorithm to be used for TROPOMI. For

example, the first sentence on page 4 could be expanded to a paragraph. This would make it easier for the reader to grasp the changes implemented for the new instrument.

Specific comments

P2, L5: Include reference for Montreal Protocol

P6, L14-18: Much of the information on total ozone is common knowledge (at least for readers interested in the paper) and could be removed. Also the following subsections on albedo, aerosols, and altitude could be reduced in lengths.

P7, L26: High latitude profiles are quite different for the northern and southern hemispheres. Are different profiles being used for the two regions in the algorithm?

P8, L2: "AI" in "taken from the TROPOMI AI L2 output" presumably stands for Aerosol Index. It is a bit surprising that cloud optical depth is estimated from the Aerosol Index. Please clarify.

P8, L12: I don't understand "flat-top response curve with a half-gaussian with a Half-Width-Half-Maximum". Please simply!

P9, L9: Please describe how values in the look up tables are interpolated (e.g., linear, spline, etc.)

P12, L2: Regarding "For the UV irradiances at selected wavelengths (i.e, 305, 310, 324, and 380 nm), we apply a wavelength-specific aerosol correction." Does the aerosol climatology by Kinne mentioned earlier provide aerosol properties at these wavelengths, and if not, how is the climatology interpolated and extrapolated to these wavelengths?

Figure 5: Please indicate the time of the satellite overpass in these figures.

Technical corrections / language

P9, L23: I don't understand "a measure of the effectiveness of radiation as regards a

specific effect". Please reword.

P11, L9 "valid the whole day" > "valid for the whole day"

---

## Referee Comment (RC2) · Anonymous Referee #2 · 21 Sep 2017

The manuscript under review provides the algorithm and background information for determining the surface ultraviolet (UV) radiation at specific wavelengths, temporal spans, and cloud conditions as observed by the TROPOMI instrument on the soon-to-be-launched Sentinel-5 Precursor. The TROPOMI surface UV values will extend the long surface UV data record originating with the Nimbus-7 TOMS instrument in 1978. This is an important data set in that it validates the purpose of the Montreal Protocol by showing increases in the surface UV radiation during the ozone depletion phase and hopefully will show stabilization and decreases in surface UV radiation as the ozone layer recovers. However, it is pointed out that changes due to climate, circulation, and aerosols may result in a different surface UV situation than there was back at the beginning of the data set. Thus the importance of continuing these observations and the data set.

General Comments:

[Figure]

P 1, L6-7. There are several occasions throughout this paper beginning here with 'OMI' and 'AC SAF' when the spelled out name of an acronym is given in parentheses following the acronym. The proper format is to provide the spelled out name first followed by the acronym in parentheses.

P1, L24. Please provide references for each of these statements: ' increased during the last decades of the 20th century', 'strongest place in the high latitudes of the southern hemisphere', 'Artic and mid-latitudes in both hemispheres have experienced UV increases'.

P2, L18. It is also important to note that the TROPOMI is a backscatter instrument and that total column ozone observations are limited to the sun-lite portions of the earth. Of course where there is no sun light there is no UV at the surface.

P3, L26. Add 'respectively' to the references as they match the European and North American regions.

P4, L14. Somewhere in the 'Heritage' section it should be noted what the Equator crossing times were for the previous surface UV observing satellites. This is important to know at what time the actual observations were made when combining data sets.

P6, L12. The effect of enhanced UV at the surface depends upon the integration time (dosage) instantaneous increases as noted have been observed, but these are not sustained over longer time integrations (minutes to hour).

P7, L11. An additional piece of information would be that this effect is in the order of 3% and that the earth is closest the sun in early January and furthest in July.

P11, L16. Where or how is the aerosol absorption optical depth determined?

P12, L4. Why is the aerosol correction applied after the erythemal and vitamin-D values are determined? Why can't the aerosol corrected UV irradiances be used to compute the erythemal and vitamin-D values?

[Figure]

P12, L15. How many scan positions are there in each swath?

P12, L32. The relationship is easy to state: UVI = 0.4 *Erythemal Dose Rate (W/sq m).

P13, L5. There is a small crescent shaped area just south of the Black Sea of high reflectance. There are mountains there. Is the high reflectance due to snow, clouds or both? This area translates into lower UV values.

P13, L6. Not to add too many more plots, but it would be nice to see observations from Jokioinen and Sodankyla on entirely clear days to show that the observations do lay on top of the green curve. Figure 5 is useful to show the potential error in computing the daily dosage from one overpass per day. However, since the overpass time and solar noon at most points along the orbit are within $\pm 2$ hours the error from overpass time to solar noon should be smaller unless rapid cloud conditions do occur.

P15, L12. Instead of telling the reader later in the paragraph that the errors were assessed at high latitudes, add that piece of information in the beginning sentence.

P15, L 27. This phrase does not make sense: 'continues to present date thanks on the OMI UV record'.
* * *

---

## Author Comment (AC1) · 4 Nov 2017

**Response to the Reviewers' Comments**

Concerning manuscript amt-2017-210, *"The TROPOMI surface UV algorithm"* by Anders V. Lindfors et al.

We have received the comments on our manuscript by two reviewers. We thank the reviewers for their constructive comments. We have considered these comments in regards of our revised manuscript. Below, we detail the comments by the reviewers together with our response to them.

**Reviewer #1**

Reviewer #1 indicates in a general remark that our manuscript is important and should be published after minor changes.

General Comment: *"My main suggestion is to also include a short paragraph summarizing the difference between the UV algorithm currently used for the processing of OMI observations and the new algorithm to be used for TROPOMI."*

Reply: We have added such a paragraph, P4/L6-12 in the revised manuscript.

Comment #1: *P2, L5: Include reference for Montreal Protocol*

Reply: Added reference as suggested.

Comment #2: *P6, L14-18. Much of the information on total ozone is common knowledge (at least for readers interested in the paper) and could be removed. Also the following subsections on albedo, aerosols, and altitude could be reduced in lengths.*

Reply: It is true that much of this is common knowledge to readers with background in UV radiation and ozone science. However, we see that potential readers of this paper go beyond that community, including users of the satellite UV product, which may have a variety of backgrounds and thus are not necessarily familiar with this subject area. Therefore, we think this information could be useful for some readers and hence have chosen to include some detail on the different factors having an influence on solar UV radiation reaching the surface. In this respect, we have chosen not make any changes to the revised manuscript.

Comment #3: *P7, L26. High latitude profiles are quite different for the northern and southern hemi-spheres. Are different profiles being used for the two regions in the algorithm?*

Reply: The TOMS V7 profile climatology does not depend on hemisphere. According to Wellemeyer et al. (1997), it would be an unwarranted complication to include a small hemispheric asymmetry in the climatology. It is worthwhile pointing out, however, that the profile climatology depends on total ozone column and hence there is some indirect dependence on hemisphere at high latitudes (see Wellemeyer et al. for details).

Comment #4: *P8, L2. "AI" in "taken from the TROPOMI AI L2 output" presumably stands for Aerosol Index. It is a bit surprising that cloud optical depth is estimated from the Aerosol Index. Please clarify.*

Reply: The cloud optical depth is estimated based on the reflectance at 354 nm (R354). R354 is used also for calculating the Aerosol Index, which is why it happens to be included in the L2 AI product, which is consequentially used as input to the UV algorithm. We have made small changes to the text here to better reflect this background.

Comment #5: *P8, L12. I don't understand "flat-top response curve with a half-gaussian with a Half-Width-Half-Maximum". Please simply!*

Reply: It is a bit unclear what the reviewer is suggesting here ("Please simply!"): perhaps simply remove this part? The sentence aims to explain what kind of slit function (spectral response) is used when simulating the reflectance at 354 nm in our radiative transfer calculations. This may be a bit of a detail, but for completeness we would like to have this information included in the text. In an attempt to make the text more understandable, we have changed the sentence to: "…combined spectral response of these 5 consecutive pixels are represented by a 0.88 nm wide (corresponding to 5 pixels 0.22 nm apart), flat-top slit function with a half-gaussian with a Half-Width-Half- Maximum of 0.27 nm at each end."

Comment #6: *P9, L9. Please describe how values in the look up tables are interpolated (e.g., linear, spline, etc.)*

Reply: LUT interpolation is performed using polynomial (Lagrangian) interpolation in multidimensional space, choosing 4 points from the surrounding space if available (resulting in 3rd degree polynomial interpolation). If the wanted point is closer to the boundary of the LUT, then 3 closest points are chosen (resulting in $2^{nd}$ degree polynomial interpolation), while for points outside the LUT space 2 points are used (resulting in linear extrapolation). This information has been added to the revised manuscript.

Comment #7: *P12, L2. Regarding "For the UV irradiances at selected wavelengths (i.e, 305, 310, 324, and 380 nm), we apply a wavelength-specific aerosol correction." Does the aerosol climatology by Kinne mentioned earlier provide aerosol properties at these wavelengths, and if not, how is the climatology interpolated and extrapolated to these wavelengths?*

Reply: Kinne's aerosol climatology utilized for the aerosol correction includes wavelengths from the UV all the way into the infrared. For the TROPOMI aerosol correction, we have used the following wavelengths: 290, 315, 345, and 380 nm. The AOD and SSA data have then been linearly interpolated to the wavelengths required by the UV algorithm, that is, 305, 310, 324, and 380 nm. Information on this has been added to the revised manuscript.

Comment #8: *Figure 5: Please indicate the time of the satellite overpass in these figures.*

Reply: The satellite overpass UVI (and its time) is now included in the figure.

Comment #9: *P9, L23: I don't understand "a measure of the effectiveness of radiation as regards a specific effect". Please reword.*

Reply: Rephrased to "In order to quantify the effectiveness of radiation with respect to a specific effect"

Comment #10: *P11, L9 "valid the whole day" > "valid for the whole day"*

Reply: Changed as suggested.

---

## Author Comment (AC2) · 4 Nov 2017

**Response to the Reviewers' Comments**

Concerning manuscript amt-2017-210, *"The TROPOMI surface UV algorithm"* by Anders V. Lindfors et al.

We have received the comments on our manuscript by two reviewers. We thank the reviewers for their constructive comments. We have considered these comments in regards of our revised manuscript. Below, we detail the comments by the reviewers together with our response to them.

**Reviewer #2**

Reviewer #2 indicates in a general remark that our paper describes the basis of an algorithm that will produce data important for continued monitoring of the surface UV radiation conditions.

Comment #1: *P 1, L6-7. There are several occasions throughout this paper beginning here with 'OMI' and 'AC SAF' when the spelled out name of an acronym is given in parentheses following the acronym. The proper format is to provide the spelled out name first followed by the acronym in parentheses.*

Reply: Corrected as suggested. On rare occasions, when an acronym had already been defined but we wanted to remind the reader of its meaning, we still use the spelled out description in parentheses (e.g., P6, L3: "found an average CMF (Cloud Modification Factor) of 0.4")

Comment #2: *P1, L24. Please provide references for each of these statements: ' increased during the last decades of the 20th century', 'strongest place in the high latitudes of the south- ern hemisphere', 'Artic and mid-latitudes in both hemispheres have experienced UV increases'.*

Reply: UNEP 2011 is the reference for all those statements. The text has been revised to make this clear to the reader.

Comment #3: *P2, L18. It is also important to note that the TROPOMI is a backscatter instrument and that total column ozone observations are limited to the sun-lite portions of the earth. Of course where there is no sun light there is no UV at the surface.*

Reply: The paragraph in question has been modified to reflect the point made by the reviewer. Furthermore, since the S5P satellite has now already been launched, the text was updated to reflect this fact, with the launch date included.

Comment #4: *P3, L26. Add 'respectively' to the references as they match the European and North American regions.*

Reply: During the revision process, an additional geostationary satellite UV algorithm came up, and we decided to refer also to that algorithm. Thus the citation at the end of the sentence is now to three papers, and therefore adding 'respectively' does not seem appropriate.

Comment #5: *P4, L14. Somewhere in the 'Heritage' section it should be noted what the Equator crossing times were for the previous surface UV observing satellites. This is important to know at what time the actual observations were made when combining data sets.*

Reply: We added information on TOMS, OMI and TROPOMI equator crossing times to the revised manuscript (P4, L19-20 in revised manuscript).

Comment #6: *P6, L12. The effect of enhanced UV at the surface depends upon the integration time (dosage) instantaneous increases as noted have been observed, but these are not sustained over longer time integrations (minutes to hour).*

Reply: Text changed to reflect this.

Comment #7: *P7, L11. An additional piece of information would be that this effect is in the order of 3% and that the earth is closest the sun in early January and furthest in July.*

Reply: We added this information to the revised manuscript.

Comment #8: *P11, L16. Where or how is the aerosol absorption optical depth determined?*

Reply: Determination of the aerosol absorption optical depth is described in the papers referred to (Arola et al., 2005; Krotkov et al., 2005). Arola et al. used an inversion algorithm based on spectral Brewer measurements, while Krotkov et al. used an inversion algorithm based on measurements of a UV-multifilter rotating shadowband radiometer (UV-MFRSR).

Comment #9: *Why is the aerosol correction applied after the erythemal and vitamin-D values are determined? Why can't the aerosol corrected UV irradiances be used to compute the erythemal and vitamin-D values?*

Reply: This has to do with the design of the algorithm. As the aerosol correction is applied as a post-correction step, in line with the studies in the literature suggesting to use this correction (Krotkov et al., 2005; Arola et al., 2005), it also means that it would be technically difficult to include the aerosol correction spectrally into the UV look-up-table before the weighted dose rates are introduced. Therefore, we have chosen to correct the erythemal and vitamin-D weighted UV dose rates according to the same scheme as used for the irradiances at single wavelengths.

Comment #10: *P12, L15. How many scan positions are there in each swath?*

Reply: The TROPOMI swath consists of 450 across-track pixels, we added this to the revised manuscript. We would like to further point out that TROPOMI is not a scanning instrument. The whole swath is measured at once using CCD detectors.

Comment #11: *P12, L32. The relationship is easy to state: UVI = 0.4 *Erythemal Dose Rate (W/sq m).*

Reply: We have added the information how to calculate the UV Index in the revised manuscript. Note, however, that the relationship is: UVI = 40 * erythemal_dose_rate [W/m2] (see WHO, 2002).

Comment #12: *P13, L5. There is a small crescent shaped area just south of the Black Sea of high reflectance. There are mountains there. Is the high reflectance due to snow, clouds or both? This area translates into lower UV values.*

Reply: This is indeed an interesting feature. Meteorological satellite data from Meteosat/SEVIRI show clouds over this region for the time corresponding to the data presented in the figure. We added a sentence on this in the revised manuscript.

Comment #13: *P13, L6. Not to add too many more plots, but it would be nice to see observations from Jokioinen and Sodankyla on entirely clear days to show that the observations do lay on top of the green curve. Figure 5 is useful to show the potential error in computing the daily dosage from one overpass per day. However, since the overpass time and solar noon at most points along the orbit are within ±2 hours the error from overpass time to solar noon should be smaller unless rapid cloud conditions do occur.*

Reply: We are restricted to the test data sets available from the TROPOMI working group. From those data, we could not find a suitable cloud-free day for FMI's UV stations. However, in order to elaborate on the point made by the reviewer, we added some text to a sentence already in the manuscript, elaborating on the agreement in Jokioinen during the cloud-free early morning and late afternoon hours ("The very morning and late afternoon hours in Jokioinen also appear cloud-free, with good agreement between the TROPOMI clear-sky UVI and that measured by the Brewer").

Comment #14: *P15, L12. Instead of telling the reader later in the paragraph that the errors were assessed at high latitudes, add that piece of information in the beginning sentence.*

Reply: Added high-latitude to the beginning as suggested.

Comment #15: *P15, L 27. This phrase does not make sense: 'continues to present date thanks on the OMI UV record'.*

Reply: Sentence changed to 'continues to present data thanks *to* the OMI UV record'.

---

## Author Response (AR2)

**Response to the Reviewers' Comments, dated 22 Dec 2017**

Concerning manuscript amt-2017-210, *"The TROPOMI surface UV algorithm"* by Anders V. Lindfors et al.

We have received the editor decision and comments on our revised manuscript by one reviewer. We thank the reviewer for his/her constructive comments that help improve the manuscript. We have made all the suggested changes to the revised version of our manuscript.

---

## Author Response (AR3)

**Response to the Editor's Comments, dated 05 Jan 2018**

Concerning manuscript amt-2017-210, *"The TROPOMI surface UV algorithm"* by Anders V. Lindfors et al.

The editor requests us to carefully consider the language before submitting the final version of the manuscript, pointing out especially a sentence on p. 3 (lines 12-13).

*Reply: We thank the editor for accepting our manuscript and for encouraging us to make a final check of the language before the manuscript will be published. We have carefully read the manuscript, including the sentence highlighted by the editor, but do not see reasons to change anything. We note also, that the other reviewer already has scrutinized our language, resulting in improvements in the previous revisions of the manuscript. Thus, we are confident that the language is of satisfactory standard for AMT.*